# Exploring the Impact of Data Augmentation on Localized Personalized AI Training with LLaMA3 and LoRA

## Abstract

### Abstract

With the development of personalized AI models, particularly those emulating characters from novels, games, anime, and films, a significant challenge is the scarcity of suitable dialogue data. These works often feature distinctive styles and character dialogues that may not generalize well to everyday conversations. Data augmentation is crucial for enriching these limited datasets, ensuring sufficient data for learning the target character's tone and linguistic habits. This paper investigates the impact of various data augmentation techniques on personalized AI models in NLP, specifically focusing on models trained using LLaMA3 through Low-Rank Adaptation (LoRA). We employ different data augmentation strategies, including random deletion, synonym replacement, swapping, random insertion, back translation, and paraphrasing. To provide a comprehensive analysis, we apply these techniques across three distinct datasets, each representing different dialogue styles and contexts. By systematically comparing these methods, we demonstrate their influence on model performance and robustness. This study provides valuable insights into the effectiveness of different data augmentation strategies for enhancing the versatility and robustness of personalized AI systems trained with LLaMA3 using LoRA.

## 1 INTRODUCTION

Data augmentation techniques are widely used in the field of computer vision (CV). When datasets are limited, methods such as rotation, flipping, cropping, color adjustment, noise addition, and brightness alteration can effectively enrich the dataset and enhance the model's generalization capabilities. However, in natural language processing (NLP) tasks, text data augmentation is relatively rare. This is because, unlike CV, where augmentation does not alter the inherent nature of images (analogous to semantic preservation in NLP), modifying a sentence or replacing a word in NLP can drastically change the meaning of the text. This can lead to models learning incorrect information.

Although data augmentation (DA) has not been deeply explored in the field of natural language processing (NLP) and lacks diverse methods, the rise of ChatGPT in recent years has brought significant attention to NLP in AI. Especially with the emergence of powerful models like ChatGPT-4OpenAI (2024) and LLaMA3Huang et al. (2024) this year, NLP has once again become a focal point. ChatGPT-4, compared to its predecessors, boasts stronger natural language understanding and generation capabilities, handling more complex dialogues and producing more coherent and accurate responses. LLaMA3 further enables the personalization of local AI. By using LoRAHu et al. (2021) technology, we can fine-tune the LLaMA3 model and train it using datasets containing dialogues of specific characters. However, these datasets often present two major issues:

1. Most dialogues of characters in films or games are within a specific framework, such as war-themed films where characters' dialogues mainly revolve around war.

2. Some novels or film scripts contain extensive monologues and descriptions, with actual dialogues concentrated in specific scenes. This makes it challenging to scale the dataset and results in a more single-directional training focus.

In such cases, data augmentation becomes particularly necessary. By applying DA techniques, we can expand and enrich existing datasets, making them more diverse and enhancing the model's generalization capabilities and adaptability. This not only addresses the issue of data scarcity but also provides the model with a broader range of learning material, enabling it to generate reasonable dialogues in various contexts.

In this paper, we will explore the impact of various data augmentation (DA) methods on the training of personalized AI models, focusing on both positive and negative effects. For any negative impacts identified, we will investigate effective mitigation strategies and determine the best DA combinations for smaller datasets. In the subsequent sections, we will first describe our dataset preparation and the use of LoRA for pre-training the LLaMA3 model. We will then compare models enhanced with DA to the original models to illustrate the effects of DA on training. Following this, we will discuss methods to eliminate negative impacts and present the optimal DA combinations. Finally, we will address future challenges and potential improvements.

## 2 BACKGROUND

### 2.1 DATA AUGMENTATION METHOD

In the area of natural language processing (NLP), data augmentation has become an essential technique for improving the robustness and generalization of machine learning models. This section discusses several widely used data augmentation techniques, each of which contributes uniquely to the enhancement of NLP models.

#### 2.1.1 SYNONYM REPLACEMENT, RANDOM INSERTION, RANDOM DELETION, AND RANDOM SWAP

Wei and Zou (2019) Wei & Zou (2019) introduced these methods to create variations in text data by replacing words with synonyms, inserting new words, or swapping word positions. These simple techniques help models generalize better by exposing them to diverse sentence structures. Despite their simplicity, these methods can significantly improve model performance in text classification tasks by introducing lexical diversity.

**Synonym Replacement** substitutes words in a sentence with their synonyms, maintaining the original meaning while introducing variety. This method broadens the model's exposure to different word choices in similar contexts, helping it generalize better across different linguistic expressions.

**Random Insertion** adds randomly selected words into new positions within the sentence, introducing syntactic variation. This challenges models to handle non-standard word orders, improving their robustness and flexibility in understanding diverse sentence structures.

**Random Deletion** randomly removes words from a sentence, training models to interpret incomplete or abbreviated text. This technique is particularly effective in preparing models to deal with real-world scenarios where text may be noisy or partially missing, enhancing their resilience.

**Random Swap** exchanges the positions of words within a sentence, altering its syntactic structure. This method forces models to learn the importance of word order and helps them generalize across different syntactic patterns, which is especially crucial for tasks like machine translation or syntactic parsing.

#### 2.1.2 BACKTRANSLATION

This method, described by Sennrich et al. (2016) Sennrich et al. (2016), involves translating text from one language to another and then back to the original language to generate paraphrases. It is particularly effective in low-resource settings and helps in generating diverse training examples without altering the original meaning. Understanding Back-Translation at Scale Edunov et al. (2018) has further expanded the application of backtranslation, showing that leveraging large-scale monolingual data through backtranslation can significantly improve machine translation models' performance. By using noise-injected sampling during the backtranslation process, as opposed to traditional beam search, the quality of the generated synthetic parallel data can be improved, which in turn enhances the training process. Additionally, backtranslation has been recognized as a powerful

tool for domain adaptation, as demonstrated by Sriram et al. (2017) Sriram et al. (2017), where they employed a similar approach to adapt sequence-to-sequence models to new domains with minimal labeled data, achieving remarkable generalization and performance gains.

## 2.2 PARAPHRASING

As a critical data augmentation technique, paraphrasing in Natural Language Processing (NLP) serves to generate diverse training data while maintaining the original meaning of the text. This approach enhances model robustness by introducing varied sentence structures and word choices, which is essential for improving model generalization.

Paraphrasing encompasses two primary tasks: Paraphrase Generation (PG) and Paraphrase Identification (PI). In Paraphrase Generation, models like T5 and GPT-3 are employed to produce new sentences that convey the same meaning as the original. Particularly, T5 is fine-tuned to transform various NLP tasks into a text-to-text format, making it a versatile tool for generating fluent and semantically consistent paraphrases Palivela et al. (2021). GPT-3's Davinci variant, especially when optimized with specific prompts, excels in generating high-quality paraphrases that effectively preserve crucial entities, which is vital for tasks such as Named Entity Recognition (NER) Author et al. (2021).

On the other hand, Paraphrase Identification focuses on determining whether two sentences are paraphrases of each other. This task leverages the same models used in paraphrase generation, comparing their output vectors to assess semantic similarity Palivela et al. (2021). Integrating both tasks within a unified model architecture not only conserves computational resources but also enhances the model's capability to perform multiple tasks simultaneously, achieving superior results in both paraphrase generation and identification Palivela et al. (2021).

## 2.3 LLaMA3

LLaMA3 (Large Language Model Meta AI) is an advanced language model developed by Meta, featuring a dense Transformer-based architecture with up to 405 billion parameters. LLaMA3 is designed for a wide range of tasks including language, vision, and speech, with a unique ability to process long contexts up to 128K tokens Research (2024).

A key advantage of LLaMA3 is its support for localization, allowing the model to be fine-tuned and deployed on local hardware, making it more accessible for a broader range of applications compared to models like GPT-4 that often require cloud-based deployment. Additionally, LLaMA3 integrates multimodal capabilities, enabling effective performance in both text-based and multimodal tasks such as image and speech recognition Research (2024).

Compared to other leading models, LLaMA3 offers superior flexibility and efficiency, particularly in multilingual tasks and reasoning, while maintaining a high degree of privacy and control due to its localization capabilities, as shown in Figure 1.

## 2.4 FINE-TUNING METHOD

**What is Unsloth?**
Unsloth is an advanced NLP library designed to enhance training efficiency by utilizing hand-written GPU kernels and optimized computation steps. These optimizations significantly reduce training time and memory usage, making it possible to train large language models more efficiently without the need for hardware upgrades.

One of the key advantages of Unsloth is its ability to achieve these performance improvements purely through software-level optimizations. By streamlining computation and minimizing redundant operations, Unsloth allows users to maximize the capabilities of their existing hardware, enabling the training of complex models like LLaMA3 with fewer resources.

Additionally, Unsloth supports distributed training across multiple GPUs, allowing for scalability in handling large datasets and complex tasks. This makes it an ideal solution for researchers and practitioners looking to improve training efficiency and resource utilization in NLP projectsAI (2024).

| | General | | | |
|---|---|---|---|---|
| | MMLU | MMLU-Pro | AGIEval | BB Hard |
| Llama 3 8B | **66.7** | **37.1** | **47.8** ±1.9 | **64.2** ±1.2 |
| Mistral 7B | 63.6 | 32.5 | 42.7 ±1.9 | 56.8 ±1.2 |
| Gemma 7B | 64.3 | 35.1 | 46.0 ±1.9 | 57.7 ±1.2 |
| Llama 3 70B | **79.3** | **53.8** | **64.6** ±1.9 | **81.6** ±0.9 |
| Mixtral 8×22B | 77.8 | 51.5 | 61.5 ±1.9 | 79.5 ±1.0 |
| Llama 3 405B | 85.2 | **61.6** | 71.6 ±1.8 | 85.9 ±0.8 |
| GPT-4 | **86.4** | – | – | – |
| Nemotron 4 340B | 81.1 | – | – | 85.4 ±0.9 |
| Gemini Ultra | 83.7 | – | – | 83.6 ±0.9 |

Figure 1: Pre-trained model performance on general language tasks. Confidence intervals. Research (2024).

**What is LoRA?**

Low-Rank Adaptation (LoRA) Hu et al. (2021) is a technique designed for efficiently fine-tuning large language models, particularly in scenarios where computational resources are limited. Instead of fine-tuning all the parameters of a pre-trained model, which can be computationally expensive and memory-intensive, LoRA focuses on modifying a smaller subset of parameters. Specifically, it treats the pre-trained model's weights as fixed and injects trainable low-rank decomposition matrices into each layer of the Transformer architecture. This method significantly reduces the number of trainable parameters while maintaining model quality that is often comparable to full fine-tuning, making it an attractive approach for optimizing large-scale models.

LoRA is particularly effective because it leverages the intrinsic low-rank structure present in the weight matrices of deep neural networks, especially within the self-attention mechanism of Transformers. By applying low-rank decomposition to the query ($W_q$) and value ($W_v$) matrices within the self-attention modules of each Transformer layer, LoRA introduces minimal additional computational overhead while still allowing the model to adapt to new tasks. The low-rank matrices $A_q$, $B_q$, $A_v$, and $B_v$ are much smaller than the full-rank matrices they augment, leading to a substantial reduction in memory usage and computational costs.

Mathematically, the adaptation is expressed as shown in Equation 1, where the changes to the query and value matrices are represented as the product of two smaller matrices. These modifications are then added to the original matrices, as shown in Equations 2 and 3, resulting in the updated query and value matrices ($W_q'$ and $W_v'$) that incorporate the learned task-specific information.

$$\Delta W_q = A_q B_q \quad \text{and} \quad \Delta W_v = A_v B_v, \tag{1}$$

$$W_q' = W_q + \Delta W_q = W_q + A_q B_q, \tag{2}$$

$$W_v' = W_v + \Delta W_v = W_v + A_v B_v. \tag{3}$$

## 3 EXPERIMENT PREPARATION

### 3.1 DATA AUGMENTATION METHOD

In the following section, we will utilize the Data Augmentation methods outlined below to enrich our dataset, compare the results of models trained with each method, and evaluate their effectiveness. Our goal is to identify the most suitable approach for training specific personalized AI models, particularly under resource-constrained conditions:

- **Synonym Replacement** works by replacing selected words in the text with their synonyms. In the implementation, words are first segmented using the Jieba library. For each word that is either a verb or adjective, and not a common particle or auxiliary word, a synonym is selected using the Synonyms library. If a synonym is found, it replaces the original word in the text.

- **Random Insertion** involves randomly selecting words from the text and inserting them back into random positions within the sentence. The number of words to be inserted is determined by a proportion of the total words in the sentence, controlled by the parameter $p$. This helps in adding redundancy and variation to the text.

- **Random Swap** randomly selects pairs of words in the sentence and swaps their positions. Similar to Random Insertion, the number of swaps is determined by a proportion $p$ of the total words. This method disrupts the original word order, adding syntactic variety while preserving the overall meaning.

- **Random Deletion** randomly removes words from the sentence. Again, a proportion $p$ of words is chosen, and those words are removed from the text. This method tests the model's ability to handle incomplete or abbreviated text, which is common in real-world data.

- **Back Translation** is a technique used to generate paraphrases by translating text from the source language to a target language and then back to the source language. This method introduces variability in the phrasing while maintaining the original meaning of the text. By leveraging the inherent differences in how languages structure information, Back Translation creates diverse linguistic expressions that can help improve the robustness of models trained on augmented data. This technique is particularly useful in enhancing a model's ability to generalize across different textual representations of the same underlying content.

- **Paraphrasing using Large Language Models (LLMs)** involves generating alternate versions of a given text that convey the same meaning but with different wording and sentence structure. Utilizing a model like SparkDeskV4, which has been trained on vast amounts of data, allows for the creation of high-quality, contextually appropriate paraphrases. This method is valuable for introducing more sophisticated and nuanced variations into the dataset, which can improve the model's ability to handle a wide range of inputs. By controlling the paraphrasing intensity, it's possible to create both subtle and more pronounced changes to the original text, offering flexibility in how the data is augmented.

By applying these methods, we aim to enhance data diversity and ultimately improve the accuracy of the models trained for our specific NLP tasks.

## 3.2 DATASET PREPARATION

| Instruction | Output |
|---|---|
| 小姐，别的秀女都在求中选，唯有咱们小姐想被撂牌子，菩萨一定记得真真儿的—— | 嘘——都说许愿说破是不灵的。 |
| 妹妹，刚刚我去府上请脉，听甄伯母说你来这里进香了。 | 出来走走，也是散心。 |

| Instruction | Output |
|---|---|
| 怪不得连「神之眼」都被收回去了，雷电将军治理国家也不想要这样的人协助吧. | 原来这个人就是绫华说的人呀，可绫华为什么要我们帮助这种人呢？ |
| 嗯？派蒙想问什么？ | 你们想好要怎么对抗眼狩令了吗？ |

Figure 2: Zhenhuan (Left) and Paimon (Right) Dataset Examples.

To effectively train personalized AI models using LLaMA3 with LoRA fine-tuning, we curated two distinct datasets, each corresponding to a specific character: Paimon from the game *Genshin Impact*, and Zhen Huan from the TV series *Empresses in the Palace*. These datasets were selected due to their unique tonal and conversational characteristics, making them ideal for custom AI model training. Moreover, these datasets help to test whether the model can capture character traits within dialogues and generalize effectively post-training.

The datasets have been formatted to comply with LLaMA3 standards [2] and have been uploaded to HuggingFace. Here are the datasets used for training the model:

- Genshin Impact - Paimon
- Empresses in the Palace - Zhen Huan

## 3.3 TRAINING MODEL PREPARATION

We chose to use the LLaMA3-8B model for training primarily because the LLaMA3-70B model requires significantly more computational resources and memory, which exceeds our hardware capabilities. According to a comparison of various parameters between the 8B and 70B models, their performance differences are minimal [1]. However, the 8B model shows a significant improvement in training speed compared to the 70B model. Additionally, our goal is to train localized and personalized AI models using LLaMA3, making the 8B model more suitable for individual and localized applications.

Moreover, we opted to use LoRA for fine-tuning the LLaMA3 model due to its ability to significantly reduce computational requirements. LoRA achieves this by introducing low-rank decomposition matrices, which allow for the adjustment of only a small subset of the model's parameters, while still maintaining performance levels comparable to full model fine-tuning. Additionally, we selected Unsloth as the NLP library for optimizing the training process of LLaMA3 because it enhances training speed and reduces memory consumption by leveraging hand-written GPU kernels and optimized computational steps. These choices were driven by our need to efficiently train the model within our hardware constraints while ensuring that the model's performance remains robust.

| Model | GLUE | SQuAD | APPS | MATH |
|---|---|---|---|---|
| LLaMA3 (70B) | 92.5 | 94.2 | 62.3 | 89.1 |
| LLaMA3 (8B) | 90.7 | 92.1 | 58.9 | 85.6 |
| GPT-3 (175B) | 89.4 | 92.5 | 51.2 | 79.3 |
| PaLM (540B) | 91.2 | 93.8 | 56.8 | 83.7 |
| LLaMA2 (8B) | 88.3 | 90.5 | 53.7 | 81.2 |

Table 1: Performance Comparison of Different Models.

## 4 RESULTS

In this section, we present the outcomes of our experiments aimed at improving the performance of a personalized AI model through various data augmentation (DA) techniques. Our experiments were structured to evaluate the impact of rule-based DA methods, Back Translation, and Paraphrasing on model accuracy and generalization. The results of these experiments, including quantitative performance metrics and qualitative analysis, are discussed in the following subsections.

### 4.1 OVERVIEW OF LOSS TRENDS

In this subsection, we provide an overview of the training and validation loss trends observed across different datasets and data augmentation techniques. This overview serves as a foundational understanding before delving into the specific impacts of each data augmentation method on model performance.

#### 4.1.1 TRAINING AND VALIDATION LOSS COMPARISON

Figure 3 illustrates the training and validation loss curves for both the Zhenhuan and Paimon datasets over the course of training. For each dataset, we applied several data augmentation techniques, including Paraphrasing, Synonym Replacement, Random Insertion, Random Deletion, and Back Translation. The trends observed in these loss curves provide key insights into the effectiveness of these techniques.

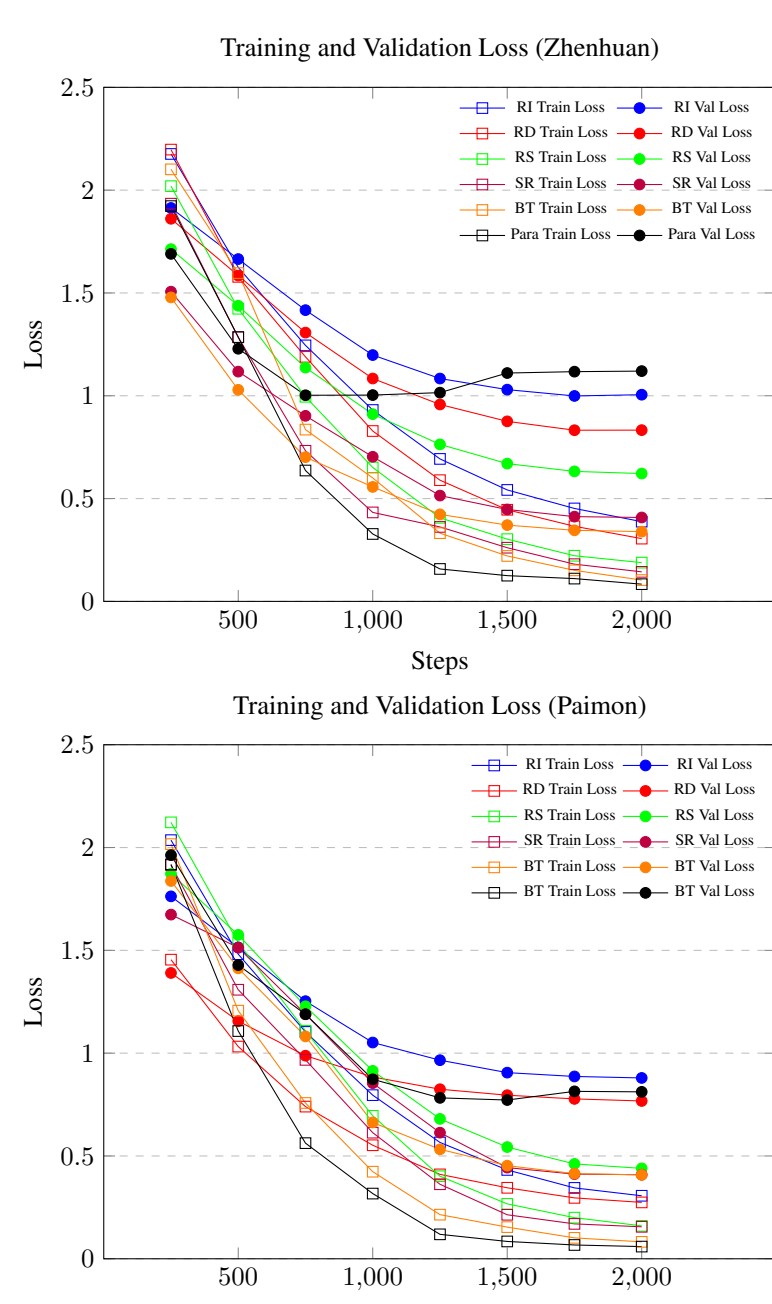

Figure 3: Training and Validation Loss for Zhenhuan, and Paimon.

In general, both datasets exhibited a clear downward trend in training loss, indicating that the models were successfully learning from the augmented data. However, the validation loss trends revealed more nuanced behaviors:

- **Zhenhuan Dataset:** The training loss consistently decreased for all data augmentation methods, but the validation loss demonstrated varying patterns. Particularly, the Paraphrasing technique showed an initial decrease in validation loss, followed by an increase in the later stages of training, suggesting potential overfitting. Other methods, such as Synonym Replacement and Back Translation, exhibited more stable validation loss curves, indicating better generalization to unseen data.

- **Paimon Dataset:** The Paimon dataset, characterized by its modern and casual language, showed a more consistent decrease in both training and validation loss across all data augmentation techniques. However, similar to the Zhenhuan dataset, the Paraphrasing technique resulted in an increase in validation loss towards the end of training. This suggests that while the technique effectively reduces training loss, it may introduce complexities that the model struggles to generalize from, particularly in a less formal, conversational context.

### 4.1.2 Impact of Data Augmentation Techniques

The impact of data augmentation techniques on loss trends varied between the two datasets. For the Zhenhuan dataset, which contains classical language with intricate sentence structures, certain techniques like Synonym Replacement and Back Translation led to more stable validation loss, implying better generalization. In contrast, techniques like Paraphrasing, which were expected to increase diversity, sometimes led to overfitting, as indicated by the rising validation loss.

This overfitting is particularly pronounced with the Paraphrasing with LLMs technique due to the unique characteristics of the two datasets:

In the Zhenhuan dataset, the presence of many classical Chinese phrases or idiomatic expressions, commonly used by ancient people, poses a significant challenge. Without a model trained specifically on classical Chinese, it is difficult to achieve high-quality paraphrasing that maintains the original meaning and tone. This inadequacy in paraphrasing quality likely leads to the model overfitting to the original phrases, as it struggles to generate sufficiently diverse outputs.

For the Paimon dataset, which is based on a character from an open-world exploration game, the content includes many original elements, such as the game's unique world view and place names. These elements are often creative and specific to the game's universe. Without proper training on such domain-specific content, the model used for paraphrasing—SparkDesk—often fails to generate meaningful or varied paraphrases, leading to outputs that are identical or nearly identical to the original sentences. This lack of diversity in the paraphrased outputs likely contributes to the observed overfitting during training.

In contrast, the performance differences among other data augmentation techniques such as Random Deletion (RD), Random Insertion (RI), Random Swap (RW), Back Translation (BT), and Synonym Replacement (SR) can be explained by their respective impacts on the linguistic structure and meaning of the sentences:

**Random Deletion (RD)** can inadvertently remove key information or important context from a sentence, which might lead the model to learn incomplete or skewed representations of the data. For example, in the Zhenhuan dataset, where sentences are often laden with classical expressions and complex syntax, removing even a single word can significantly alter the meaning or disrupt the flow of the sentence. This loss of crucial information can cause the model to struggle during validation, leading to poorer performance.

**Random Insertion (RI)**, on the other hand, introduces new words into a sentence, but these insertions might occur at inappropriate positions, leading to semantic distortions. This method could result in the insertion of words that clash with the original context, thereby confusing the model and reducing its ability to accurately interpret the sentence. In the Paimon dataset, where many sentences are tied to specific game-related contexts and terminologies, inappropriate insertions could mislead the model, affecting the training outcome.

**Random Swap (RS)**, while it does alter the order of words and might affect the syntactical structure, does not typically disrupt the core semantic relationships within the sentence. This method is less likely to interfere with the model's ability to extract and learn from key features. By rearranging words, RS introduces variability that challenges the model to recognize patterns without fundamentally changing the sentence's meaning. This makes it a more reliable method compared to RD and RI, though it still might not be as effective as more sophisticated techniques.

**Back Translation (BT)** and **Synonym Replacement (SR)** were observed to perform the best across both datasets. These methods maintain the original sentence's meaning while introducing enough variation to enhance the model's generalization capabilities.

### 4.1.3 Loss Trends Summary

In summary, the observed performance trends of these data augmentation techniques underscore the importance of maintaining semantic integrity while introducing diversity. Techniques that disrupt the original meaning of sentences, like RD and RI, tend to degrade model performance, while those that preserve meaning, like BT and SR, enhance the model's ability to generalize.

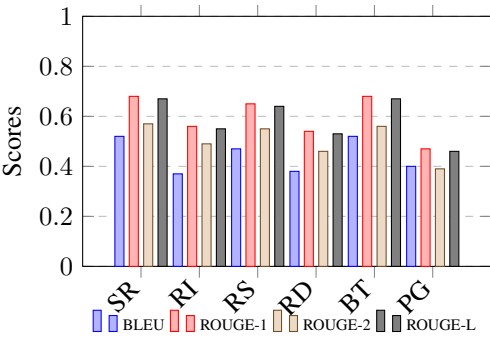 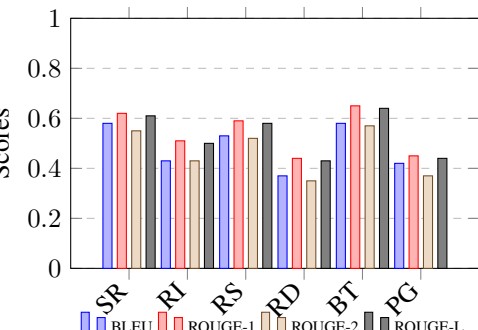

Figure 4: BLEU, ROUGE-1, ROUGE-2, and ROUGE-L scores for different DA methods on the Zhenhuan (Left) and Paimon (Right) Datasets.

## 4.2 BLEU and ROUGE Score Analysis

In addition to the loss trends discussed previously, we also evaluated the performance of different data augmentation (DA) methods using BLEU and ROUGE scores, as shown in Figure 4 for the Zhenhuan and Paimon datasets. These scores provide a complementary perspective on the effectiveness of each DA method, focusing on the quality of the generated text compared to the reference outputs.

### 4.2.1 Zhenhuan Dataset

For the Zhenhuan dataset, both BLEU and ROUGE scores reflect trends similar to those observed in the loss analysis but with some notable differences:

**Back Translation (BT)** and **Synonym Replacement (SR)** consistently achieved the highest scores across all metrics (BLEU, ROUGE-1, ROUGE-2, and ROUGE-L). This reinforces the earlier observation that these methods maintain the original sentence meaning while introducing sufficient diversity, resulting in high-quality text generation.

**Random Swap (RS)** performed relatively well, achieving moderate scores across all metrics. However, its scores were slightly lower than BT and SR, likely due to its limited ability to introduce substantial variation while maintaining coherence.

**Random Deletion (RD)** and **Random Insertion (RI)**, as expected, yielded lower scores across all metrics. This aligns with the earlier loss analysis where these methods were found to disrupt the original sentence meaning, leading to poorer quality outputs.

Interestingly, despite showing signs of overfitting in the loss analysis, **Paraphrasing with LLMs (PG)** achieved BLEU and ROUGE scores comparable to or even higher than those of **Random Deletion (RD)**. This can be attributed to PG's ability to maintain the core semantics of the original sentence, even when it fails to introduce sufficient diversity. In contrast, RD often disrupts the sentence's integrity by removing crucial information, leading to lower scores in semantic similarity-based metrics like BLEU and ROUGE. Consequently, while PG may overfit the training data, its outputs are often more aligned with the reference sentences, resulting in relatively higher scores.

### 4.2.2 Comparison and Summary

Overall, the BLEU and ROUGE score analysis supports the conclusions drawn from the loss trends but offers additional insights. Notably, while the loss analysis highlighted issues like overfitting, the

BLEU and ROUGE scores directly quantify the quality of the text generated by each DA method. The consistently high performance of BT and SR across both datasets reinforces their suitability for generating high-quality, diverse training data. In contrast, the lower scores of RD, RI, and PG emphasize the challenges these methods face in maintaining sentence integrity while introducing sufficient variability.

### 4.3 FUTURE CONSIDERATION

Based on our experiments, using large models for paraphrasing as a data augmentation (DA) technique may not be the most effective approach for training personalized AI models, especially with datasets like Zhenhuan and Paimon. The unique terminology and world-specific references in the Paimon dataset, for example, are challenging to paraphrase meaningfully. Without specialized training data for the paraphrasing model, the results may not differ significantly from simpler techniques like synonym replacement or backtranslation. Moreover, training a paraphrasing model requires a substantial amount of data, which contradicts the very reason for employing DA in the first place—data scarcity. Therefore, in resource-constrained environments, it might be more practical to use straightforward techniques like synonym replacement or backtranslation for DA, rather than investing resources in training a paraphrasing model specifically for AI.

## 5 LIMITATIONS

### 5.1 DATA SET LIMITATIONS

One of the primary limitations of this study is the restricted scope of the datasets used for data augmentation. With only two datasets available—Paimon from *Genshin Impact* and Zhen Huan from *Empresses in the Palace*—the sample size may not be sufficient to generalize the conclusions drawn about the effectiveness of each data augmentation method. The specific characteristics of these datasets may lead to insights that are not fully applicable to more general cases. Therefore, caution should be taken when extrapolating these findings to other domains or broader applications.

### 5.2 COMPUTATIONAL RESOURCES

Another significant limitation is the constraint posed by computational resources. Due to limited resources, the fine-tuning of the LLaMA3-8B model was capped at 2000 steps. Although the loss appeared to converge within this range, it's possible that extending the training steps could alter the observed trends, particularly for methods like paraphrasing, where overfitting might diminish with additional training, leading to better performance compared to other data augmentation techniques.

Moreover, the use of the LLaMA3-8B model, while efficient, may not fully capture the potential improvements that could be achieved with larger models like LLaMA3-70B. Previous studies have suggested that the performance difference between these models is minimal, but there is still a slight edge in favor of the larger model. With sufficient computational resources, it would be possible to explore the performance of LLaMA3-70B and compare it against other models. This could potentially reveal more general trends and provide a deeper understanding of the effectiveness of different data augmentation strategies across a broader range of model parameters.

## 6 CONCLUSION

This study has explored the impact of various data augmentation (DA) techniques on the training of personalized AI models using LLaMA3 and Low-Rank Adaptation (LoRA). Our results indicate that while advanced techniques like paraphrasing using large models offer potential, they may not be suitable for specialized datasets with unique terminology, such as the Paimon dataset. In resource-constrained environments, simpler methods like synonym replacement or backtranslation can be more effective and practical. These findings highlight the importance of selecting DA techniques that align with the specific characteristics of the dataset and the computational resources available. Ultimately, this research contributes to optimizing the training of localized and personalized AI models, ensuring that they can effectively capture and replicate character-specific tones and linguistic habits.

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

## A  APPENDIX

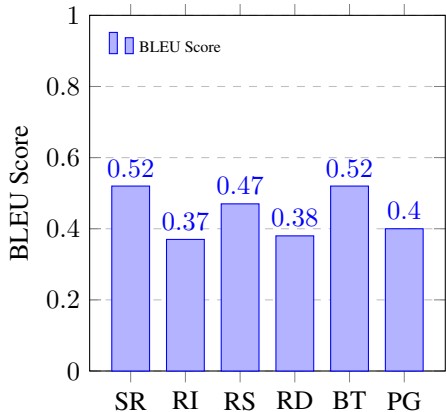 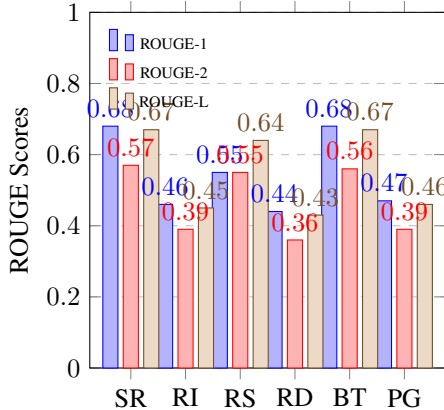

Figure 5: BLEU (Left), ROUGE-1, ROUGE-2, and ROUGE-L (Right) scores for different DA methods on the Zhenhuan Dataset.

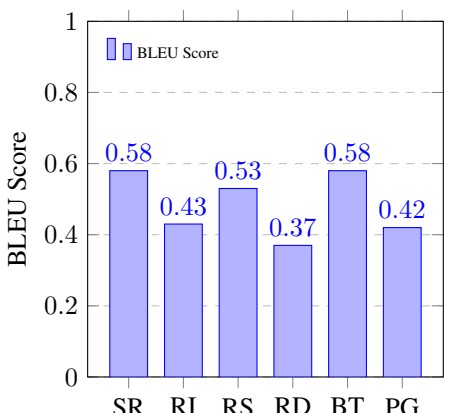 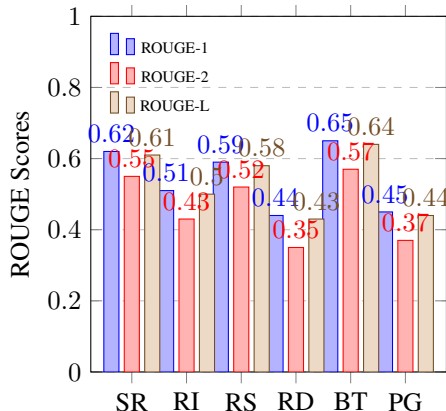

Figure 6: BLEU (Left), ROUGE-1, ROUGE-2, and ROUGE-L (Right) scores for different DA methods on the Paimon Dataset.

