# OpenReview forum: "EXPLORING THE IMPACT OF DATA AUGMENTATION ON LOCALIZED PERSONALIZED AI TRAINING WITH LLAMA3 AND LORA"
_ICLR.cc/2025/Conference — Submitted to ICLR 2025_

### Official Review · Reviewer_feda · 2024-10-23

**Soundness:** 1
**Presentation:** 1
**Contribution:** 1
**Rating:** 1
**Confidence:** 5

**Summary:**

This paper studies the use of text based data augmentations techniques for learning models which can role play as specific characters in diverse scenarios. The paper compares 6 data augmentations strategies on two characters whose content they sample from games as the basis of training Llama 3 using LoRA.

**Strengths:**

- At present, the paper does not offer any clear value to the community. The problem of data gathering and evaluating LLMs ability to roleplay is an important challenge, but the present work fails to engage with the broader literature both on these particular models and more specifically in this research space nor to offer carefully controlled experiments which answer questions from the wider field.

**Weaknesses:**

- The methods studied don't address the core challenges the authors themselves present. While the authors highlight issues that most training data used for this task occurs in a particular thematic framework as a limitation, the data augmentations they use, which are all methods which have been carefully studied in existing work but are not attributed, only perform local word and structure level permutations on data.
- The paper, as written now, spends much more time explaining (somewhat tangential) background than seems necessary. The concrete experimental setup doesn't begin until almost halfway through the work.
\
\
For example, the section on Unsloth in section 2.4 and the inclusion of external model evaluations on benchmarks that are unrelated to personalization in *both* Figure and Table 1 have little to do with the core research questions. These sections distract from the narrative of the paper and should be moved to the appendix to make room for more substantive experiments on the primary topic.
- The work primarily presents a study on just two characters from role-playing that the paper both introduces itself and compares minimally to any external baselines. It is poorly contextualized in the field without evaluations on datasets from the broader community or comparisons to even simple baselines like prompting the model to roleplay.
- There are many signs that the work was minimally refined before submission. Despite having only 11 citations , in these minimal citations there are obvious errors such as listing "Journal Name" as the publication venue and "Unknown Author et al" as the authors (the work in reference has both publicly known authors and a venue of publication: https://openreview.net/forum?id=rc2h1h89aDi). The figures have mislabeled legends (in figure 3 `BT Loss` is repeated twice) and there are typos where content is duplicated at the start of paragraphs (Line 79: "Wei and Zou (2019) Wei & Zou (2019) introduced")

**Questions:**

- Why were the data augmentations techniques not compared to finetuning *without* any data augmentations? This seems like the simplest baseline, so it's unclear why this was not evaluated.
- These BLEU and ROUGE scores seems extremely low. Are there any examples of model outputs or understanding of why they are so low? As is, these scores are likely to be reached by a model that just outputs any Chinese tokens rather than one that is personalized.
- Similarly, the decreasing loss shown could simply be a function of the model learning to always output Chinese, rather than the other languages it can output. How did you evaluate the forgetting effects of your method?
- In what context are the models evaluated on these new datasets? Is it testing whether the model is able to repeat the phrases a character speaks in it's existing context really a sound way of evaluating whether it is well suited to role-playing for downstream use?

---

### Official Review · Reviewer_RMLv · 2024-11-02

**Soundness:** 1
**Presentation:** 1
**Contribution:** 1
**Rating:** 1
**Confidence:** 5

**Summary:**

The paper explores different data augmentation strategies (i.e., random deletion, synonym replacement, swapping, random insertion, back translation, and paraphrasing) to train personalized AI models. The model is evaluate by using LoRA and two dataset proposed by the authors.

**Strengths:**

None

**Weaknesses:**

- Overall lack of clarity. It is unclear to me what is the major claim of the paper. The paper mention several techniques on data augmentation methods, but none of the it's explained in details nor formally. The results are also mysterious, leaving out most of the details.
- The results on the paper are non-sensical, the plots are meaningless, showing no particular trends.
- The paper contains screenshots from other papers (llama3), without a clear citations.

**Questions:**

None

---

### Official Review · Reviewer_2yEK · 2024-11-04

**Soundness:** 2
**Presentation:** 1
**Contribution:** 2
**Rating:** 1
**Confidence:** 5

**Summary:**

This paper investigates the effectiveness of various data augmentation techniques for training localized, personalized AI models using LLaMA3, fine-tuned through LoRA. The study employs methods like random deletion, synonym replacement, backtranslation, and paraphrasing to enhance the quality and robustness of personalized dialogue models trained on character-specific data. Experiments are conducted on two datasets with distinct linguistic styles, aiming to provide a detailed comparison of augmentation techniques in low-resource, domain-specific training scenarios.

I find that while this paper tackles a relevant problem and provides an insightful comparison of augmentation techniques, its contributions remain limited. The paper needs a more novel methodological approach or a broader set of experiments that highlight its findings' relevance to a wider range of personalized AI training scenarios. Therefore, I unfortunately cannot recommend acceptance for this paper; however, I have included some suggestions for improvements in my comments below.

**Strengths:**

1. **Clear and Relevant Problem Statement**: The scarcity of personalized dialogue data is an issue of practical importance for character-based AI models. The study addresses this with a focus on data augmentation, an area less explored in NLP.
2. **Multiple Metrics Used**: The paper evaluates on both training and validation loss, as well as BLEU and ROUGE scores, providing valuable insights for optimizing dialogue data augmentation.
3. **Resource-Conscious Approach**: The authors' choice of LoRA for fine-tuning and Unsloth for training efficiency is thoughtful and appropriate for low-resource environments, demonstrating an understanding of practical constraints in the field.

**Weaknesses:**

1. **Incremental Contributions**: While the topic of data augmentation in NLP is valuable, the paper’s contributions appear largely incremental. Many of the techniques explored (e.g., synonym replacement, backtranslation) are well-known and widely used. The paper could benefit from exploring more novel augmentation techniques tailored to character-based dialogue data's nuances. Additionally, *about half of the paper is describing past techniques* like EDA, Paraphrasing, LoRA, etc (specifically, Section 2, and 3), and there is nearly not enough content for this paper to be considered an impactful contribution (figure 3 has too little information to be taking a whole page and Table 1 font seems overtly big as if just to fill some space).
2. **Limited Dataset Scope**: Only two datasets are used, both of which are highly specific in context (e.g., Paimon from *Genshin Impact*). This narrow focus raises questions about the generalizability of the findings to other personalized dialogue systems or broader NLP applications. Including a wider variety of datasets or character types would add robustness to the conclusions.
3. **Paraphrasing Overfitting Concerns**: The paper notes overfitting issues with the paraphrasing technique but does not fully explore ways to mitigate this. Although the authors acknowledge the limitations of paraphrasing with large models in domain-specific contexts, a more thorough investigation into alternative paraphrasing techniques or domain adaptation methods would strengthen the study.
4. **Issues With the References**: a) There are some invalid entries in the references, such as, "Unknown Author" and b) The reference list is too short. I suggest the authors to conduct a more thorough review of the current literature.

### Suggestions

1. **Broaden Dataset Variety**: I think the authors are already aware of this problem, but including datasets that represent a wider range of conversational styles or character personalities would improve the generalizability of the results. Alternatively, conducting a detailed analysis of *why* certain augmentation techniques are particularly suited or unsuited to the specific characteristics of these datasets would be more interesting.
2. **Investigate Advanced Augmentation Techniques**: Consider exploring novel or advanced data augmentation techniques that could better align with the semantic and contextual requirements of character dialogue, such as sentiment-aware or context-sensitive paraphrasing. Additionally, using domain-adaptive techniques for paraphrasing may mitigate overfitting while still generating diverse examples. There is also a huge body of work on using off-the-shelf LLMs for data augmentation that the authors should explore.
3. **Refine Analysis of Computational Constraints**: Given the choice to work with the smaller LLaMA3-8B model, it would be beneficial to address the specific trade-offs involved more explicitly. This discussion could include any adjustments made to balance memory efficiency with model performance and how this may limit model generalization capabilities.
4. **Improve Paper Organization**: We don't need to describe past methods like EDA, LoRA, etc. in so much detail. Instead, I would suggest focusing on including more strong baselines (like LLM-based data augmentation).

**Questions:**

N/A

---

### Official Review · Reviewer_AruV · 2024-11-04

**Soundness:** 1
**Presentation:** 3
**Contribution:** 1
**Rating:** 3
**Confidence:** 4

**Summary:**

The paper investigates the relevance of data augmentation techniques for the training of personalized dialogue agents. It gives an overview of existing text augmentation methods, of existing open-weights LLMs and of low-rank adaptation techniques. It introduces a custom dataset based on a TV series and a video game, and a task consisting of enacting a specific character within these environments.

**Strengths:**

# Originality

The proposed approach consists of low-rank fine-tuning applied on a training dialogue dataset with data augmentation. The approach in itself is not particularly novel, and its presentation would benefit from a more thorough literature review.

The introduced dataset is novel, and its elaboration would benefit from a more detailed presentation.

# Quality

Only one set of experiments has been run, using one available open-weight model and the dataset that the paper introduces. This is insufficient to get to sound conclusions as to which data augmentation techniques are better in general for the task of dialogue modeling.

# Clarity

The paper is easy to follow, but misses key details in how the dataset has been constructed: the paper should mention where the data comes from, how it was processed, what are the key characteristics (# of episodes, of tokens, of characters...), how the training/validation split was done...

# Significance

The understanding of which data augmentation techniques are most relevant in order to adapt LLMs to small datasets is of prime importance for the field. However, it seems that the proposed method and the results do not give very strong indications of winning strategies.

**Weaknesses:**

Regarding the proposed dataset:
* It is not clear where the training data comes from and how it was curated
* It is not clear how the proposed dataset relates to existing work, and which specific problems it aims at solving
* There is no measurement of training data leakage. Given their popularity, it seems likely that these TV shows are quite prevalent in LLM training data

Regarding the fine-tuning results:
* There is no comparison with a baseline that does not use data augmentation
* There is no comparison to any few-shot prompting scenario
* It is not clear what the fine-tuning for a specific character consists of: how is the autoregressive loss computed when multiple characters are talking?

Regarding the paper presentation:
* The paper spends several pages presenting topics that are not directly relevant for the subject at hand. While understanding low-rank fine-tuning is a topic of importance, it is not particularly relevant to spend 1 page on it, at the expense of experimental details and results.
* The abstract mentions 3 different datasets, but the results cover only two of them.

**Questions:**

* Could you check the supplementary material? The content provided as supplementary material is not a zipfile and can't be decompressed by zip utility.

* How well would prompt-based solutions fare in this setup?

* Would the results still hold when used on existing dialogue datasets?

---

### Meta-Review · Area_Chair_1JGS · 2024-12-16

**Metareview:**

The authors investigate current data augmentation methods for training personalized / character-based models and also propose a new dataset based on a TV series and a video game. While the reviewers appreciate the relevance and timeliness of this work, they raise significant concerns such as the limited contribution of the paper, limited scope of evaluation, clarity of presentation, and insufficient knowledge of related work. Given the above, I do not believe this work is mature for publication.

**Additional Comments On Reviewer Discussion:**

No discussions held between authors and reviewers (the authors did not provide a response).

---

### Decision · Program_Chairs · 2025-01-22

Reject